# Data Analysis of Shipment for Textiles and Apparel from Logistics Warehouse to Store Considering Disposal Risk

**Rina Tanaka [1], Aya Ishigaki [1,*], Tomomichi Suzuki [1], Masato Hamada [2] and Wataru Kawai [2]**

[1]   Industrial Administration, Tokyo University of Science, Noda, Chiba 278-8510, Japan;
7418519@ed.tus.ac.jp (R.T.); szk@rs.tus.ac.jp (T.S.)

[2]   Data-Chef Co., Ltd., Koto-ku, Tokyo 135-0004, Japan; m.hamada@data-chef.co.jp (M.H.);
w.kawai@data-chef.co.jp (W.K.)

*   Correspondence: ishigaki@rs.noda.tus.ac.jp; Tel.: +81-4-7124-1501

**Abstract:** Given the rapid diversification of products in the textile and apparel industry, manufacturers face significant new challenges in production. The life cycle of apparel products has contracted and is now, generally, a several-week season, during which time a majority of products are supposed to be sold. Products that do not sell well may be sold at a price lower than the fixed price, and products that do not sell at all within the sales period may eventually become forced disposal. This creates long-term management and environmental problems. In practice, shipping personnel determine when to ship products to stores after reviewing product sales information. However, they may not schedule or structure these shipments properly because they cannot effectively monitor sales for a large number of products. In this paper, shipment is considered to reduce the risk of product disposal on the premise of selling at a fixed price. Although shipment quantities are determined by various factors, we only consider the change in inventory at the logistics warehouse, since it is difficult to incorporate all factors into the analysis. From cluster analysis, it is found that shipping personnel should recognize a policy to sell products gradually over time. Furthermore, to reduce the risk of disposal, we forecast the inventory from conditional probability and are able to extract products out of a standard grouping using past data.

**Keywords:** apparel products; supply chain management; quick response; clustering; forecasting; sale at fixed price

## 1. Introduction

Environmental problems such as global warming and water shortages along with the expansion of social disparities from globalization not only remain unsolved in the world, they continue to escalate. Thus, the need for companies to play significant roles in shaping sustainable societies is also increasing [1]. A society can be considered sustainable from three perspectives—environmental, societal, and economic [2]; these are often referred to as the triple bottom line or 3BL [3]. Companies must consider the impact of their business activities on the 3BL and establish long-term corporate strategies to minimize adverse impacts in any of these dimensions. By employing this perspective in supply chain management, in particular, companies are more likely to achieve sustainability [4], which in turn can enhance a company's competitive advantage [5]. Even in the textile and apparel industry—which is subject to the vagaries of fashion and, thus, characterized by short product life cycles—companies are focusing more on sustainability as globalization progresses [6]. In this industry, the emphasis in sustainable supply chain management is placed on effective risk management.

In a textile and apparel supply chain, a company's factories manufacture products according to a designated production plan. Afterwards, via logistics warehouses, the company ships finished products to stores. At the start of a sales period, a quantity of products is made available at stores based on the production plan. At the end of the sales period, it is hoped that inventories of stores and the logistics warehouse will have harmonized with the production plan as well as with customer demand. In practice, however, customer demand often deviates from what is anticipated in the production plan, and a surplus or shortage results. This is because of the long lead times before a sales period [7] and because apparel products are seasonal, that is, usually only one season. Further, in the textile and apparel industry, the logistics warehouses serve as stock points, and therefore, decisions about shipments to logistics warehouses are crucial [8].

In recent years, as customer needs and preferences have proliferated, textile and apparel products have been made available in an ever-widening range of colors and sizes [9]. Moreover, an increasing number of manufacturers also sell more general apparel products—such as bags, shoes, and accessories—in addition to clothes [10]. Because of these trends, leading to modifications in the design of apparel products every season, demand uncertainty in the industry is large compared to that for industries with more stable products, such as office supplies, or furniture. In other words, every season there are new apparel products with very short life cycles [9]. Therefore, it is difficult to forecast when and which products will be sold at any given time [11].

Many factors are at play in textile and apparel supply chains. When shipping products from the production plants to the logistics warehouse, timing and quantities are determined in advance. On the other hand, when shipping from the logistics warehouse to stores, timing and quantities can be modified in accordance with customer demand by taking into account actual sales information. The earlier that shipments are sent from the logistics warehouse, the higher is the possibility that inventory at the stores will become excessive, such that unsold products will eventually be returned to the logistics warehouse and discarded. Inventory space in a store may also be quite limited, resulting in the need to return or dispose of excess product. In contrast, when shipments from the logistics warehouse are delayed, stores can experience shortages and, as a result, lose opportunities for sales. In addition, because of long lead times before sales seasons, trends may have already have shifted by the time the season arrives, with the result that product designs will have become unsuitable for selling.

Shipping personnel are responsible for controlling inventory by determining shipments (timing and quantities) from the logistics warehouse to stores. Normally, these personnel rely on signals from stores when deciding if additional shipments are necessary [12]. Because of the range of products and frequency of shipments, however, it is difficult for them to efficiently and accurately track all sales information. These shortcomings can result in excess product, the proper disposal of which has become a significant environmental problem. The textile and apparel industry is the second largest industry in the world; as such, it can have an enormous impact on the global environment [13], particularly since waste is generated at every stage of in the apparel manufacturing process [14].

In sum, the disposal of excess product within the apparel industry is an increasing threat to the environment [15]. To minimize the quantity of unsold product, manufacturers should more carefully consider customer demand when making production decisions. In this way, they can fulfill their role in helping to build a sustainable society, at least from an environmental perspective. From an economic perspective, manufacturers face another related challenge: anticipating a disposal problem, they may significantly lower the price of products to try to sell more product and, hence, mitigate the challenge of disposal. An increase in the number of unsold products at the end of a sales cycle, then, would lead to an increase in the number of products whose price was discounted by the company. If this cycle were to continue, profits would inevitably decline and sustaining the business over the long term would become more difficult. To prevent problems such as this in the economic dimension, companies would need to increase or at least maintain the number of products sold at a fixed price.

On the premise of selling at a fixed price, we consider the timing of shipping as a method to reduce waste risk due to unsold inventory in this study. Initially, the sales scenario of the products that the shipping personnel need to monitor is restricted. Next, in consideration of the need to sell these products at a fixed price, personnel forecast the inventory ratio in the sales period and consider shipping timing in order to reduce the disposal risk.

## 2. Literature Review

Unsustainable and unregulated production and consumption are contributing to the degradation of the global environment [16]. Proper monitoring and analysis in the textile and apparel supply chain is necessary to minimize its environmental impact [17]. A literature review on the apparel supply chain is as follows.

The sustainability of the textile and apparel supply chain has two aspects: environment sustainability such as reduction of unsold products and economic sustainability that improves return on investment (ROI) [18]. With environment sustainability, manufacturers are required to manage resource procurement, inventory, waste treatment, and transport efficiently. On the other hand, with economic sustainability, manufacturers are required to increase the number of products sold at regular price and have large sales. Caniato et al. investigated large and small companies on the problem of environment sustainability in fashion supply chain [19]. As a result, large companies tend to focus more on improving products and processes, and small and medium companies tend to rebuild supply chains. Choi and Chiu proposed the mean-downside-risk (MDR) and mean-variance (MV) newsvendor model considering both environment sustainability and economic sustainability [20].

Brun and Castelli explored specific factors or product characteristics that promote competition in the fashion industry [21]. Čiarnienė and Vienažindienė identified key features of the modern fashion industry and developed a time efficient supply chain model [22]. Mehrjoo and Pasek described the structure of the apparel supply chain using system dynamics and quantitatively evaluated its risk [23]. Patil et al. developed a mathematical model to evaluate procurement, shipping, and markdown plans for new short-term life cycle products considering discounts associated with large-scale purchase and bulk transport [24].

In order for the shipping personnel to recognize when to send additional shipments, they rely on signals that indicate when products have been sold at stores. This system is known as Quick Response (QR), and it has demand-driven characteristics. Quick Response reduces lead time and allows shippers to quickly respond to market needs [25]. It was first implemented in the US apparel industry in the middle 1980s as a strategy for global response [26] and, since then, its use has spread to other industries. It was designed to be an inventory management strategy that was based on sharing information among supply chain agents [27]. By using a QR strategy, lead time is shorter because the shipping personnel can better forecast customer demand and subsequently modify the production plan [25]. In basic QR, a retailer sends point of sale (POS) data to its supplier, who then uses this information to improve demand forecasting [28]. Tsukagoshi et al. proposed using a QR inventory management strategy of introducing a monitoring period prior to the normal sales period in cases for which replenishment lead time is short compared to the sales period. They found that this approach reduced total shortage costs [29]. Tanaka et al. analyzed shipping data on the premise of selling at a fixed price [8]. They considered the risk of running out of stock but not the risk of disposal of unsold products.

In the apparel market, sales forecasting plays an increasingly important role in decision support systems for market competition and globalization [30]. Thomassey explained specific constraints and needs related to supply chain and sales forecasts in the textile apparel industry [31]. In a large set of SKUs and two consecutive sales seasons, Mostard et al. compared the accuracy of three quantitative forecasting methods based on advance demand information [11]. Fan et al. proposed a prediction model combining the Bass/Norton model and the sentiment analysis method [32].

## 3. Status of Products

In general, a manufacturer stores new products in one logistics warehouse for each supply chain. Also, a range of products can be shipped from one logistics warehouse to multiple stores, but the specific products and quantities shipped depend on the unique properties of stores, such as clientele, sales floor space, and geographic location or climate. If one attempts to consider all the characteristics of stores in detail, the scope of the data becomes very large. For this reason, we analyze the composition of shipping from one logistics warehouse to one store. Hence, in this paper, the textile and apparel supply chain is considered as shown in the Figure 1. This supply chain includes multiple factories, one logistics warehouse, one store, and many customers. Products arrive in the logistics warehouse from multiple factories, and then the shipping personnel ship products that have arrived in the logistics warehouse to one store.

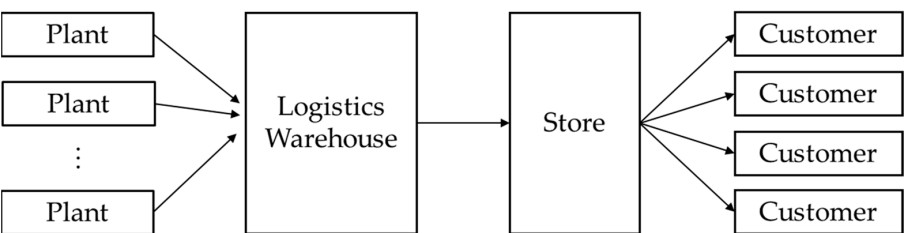

**Figure 1.** Conceptual Diagram of Supply Chain Flow.

Then, we consider inventory change in our logistics warehouse. Figure 2 shows changes in the percentages of the inventories of products in an actual logistics warehouse. As shown in Figure 2, there are products with inventory ratios of zero in the logistics warehouse at nine weeks, while some products remain in high proportions. Figure 3 shows a histogram of the inventory ratios at the ninth week. The ratio is less than 0.10 for most products, but there are still products with considerable inventory remaining. There are two reasons why large inventories for a product could persist. The first is overproduction, that is, a discrepancy between scheduled and actual sales of product. As a countermeasure, there is the method of discounting and selling inventory or selling it through another channel, such as an electronic commerce (EC) site. The second potential cause is that shipments have been timed poorly. The shipping personnel are the ones who determine the timing and quantities of products to be shipped based on sales information. However, due to the challenge of managing a large number of products, personnel may overlook a signal that a product has sold at its fixed price. If the number of unsold fixed-price products increases, there will be an increase in inventory in need of disposal at the end of the sales period. After considering sustainability, firms should increase the number of products sold at a fixed price as much as possible.

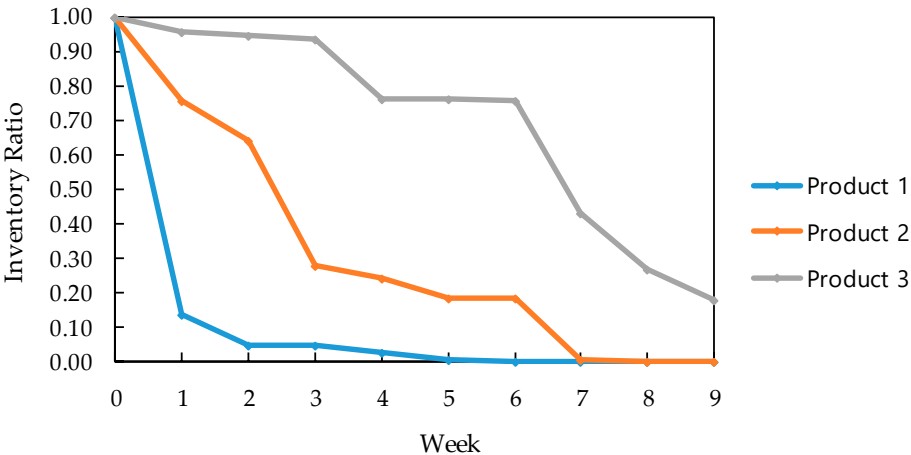

**Figure 2.** Example of Inventory Ratio Change.

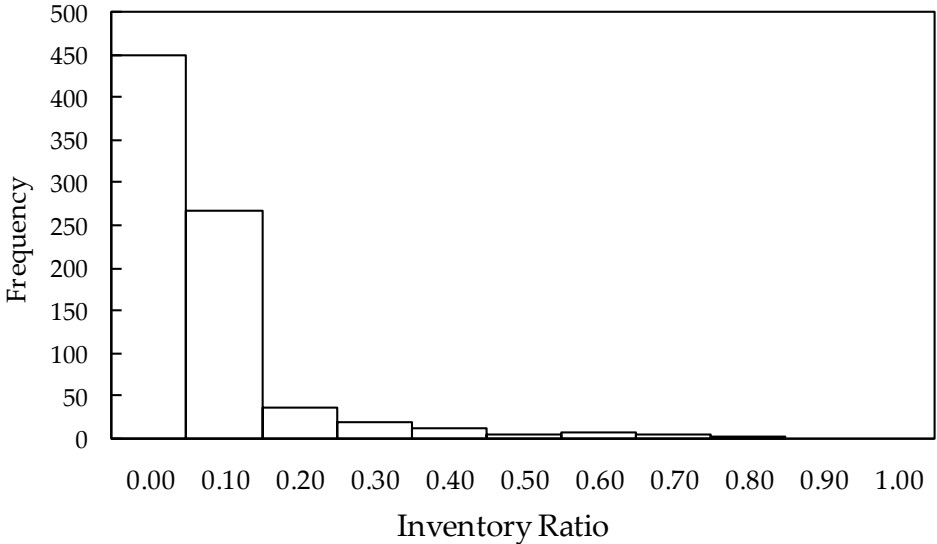

**Figure 3.** Histogram of Inventory Ratio in Ninth Week.

## 4. Methods

In this paper, we assume the supply chain as shown in the Figure 1. Since the sales cycle of the textile and apparel supply chain is one week [33], we perform our analysis using weekly data. The data used in this paper is collected in the logistics warehouse.

### 4.1. Assumptions, Parameters, and Variables

In this paper, the following assumptions, parameters, and variables are used. The parameters and variables are described in subsequent sections.

**Assumptions:**

- Products are sold in the same period
- The total arrival quantity of each product is known
- Product movement among stores is not considered
- The period in which products are sold at the fixed price is nine weeks
- Since products are sold up for nine weeks, the sales period is nine weeks
- If the product did not sell for nine weeks, the product is discarded

**Parameters and Variables:**

$N$: number of products
$T$: sales period [weeks]
$t$: elapsed periods [weeks] ($t = 1, \ldots , T$)
$i$: product number ($i = 1, \ldots , N$)
$d_i$ ($t$): shipping amount of product $i$ in week $t$
$D_i$: total delivery amount of product $i$
$u_i$ ($t$): shipping rate in week $t$ of product $i$
$K$: number of clusters
$T'$: investigated sales period
$I_{i,t}$: inventory ratio of product $i$ in week $t$
$X_i$: random variable representing the realized inventory ratio of product $i$
$Y_i$: random variable representing the predicted inventory ratio of product $i$
$\mu'$: conditional expectation

$\mu_X$: average of $X_i$

$\mu_Y$: average of $Y_i$

$\rho$: correlation coefficient

$\sigma_x$: variance of $X_i$

$\sigma_Y$: variance of $Y_i$

$\sigma'$: variance in conditional probability

### 4.2. Decision-Making Process for Shipping Personnel

Based on the above parameters and variables, Figure 4 shows the decision-making process of shipping personnel. It has seven steps. This paper proposes methods for supporting shipping personnel who are in charge of each decision-making process.

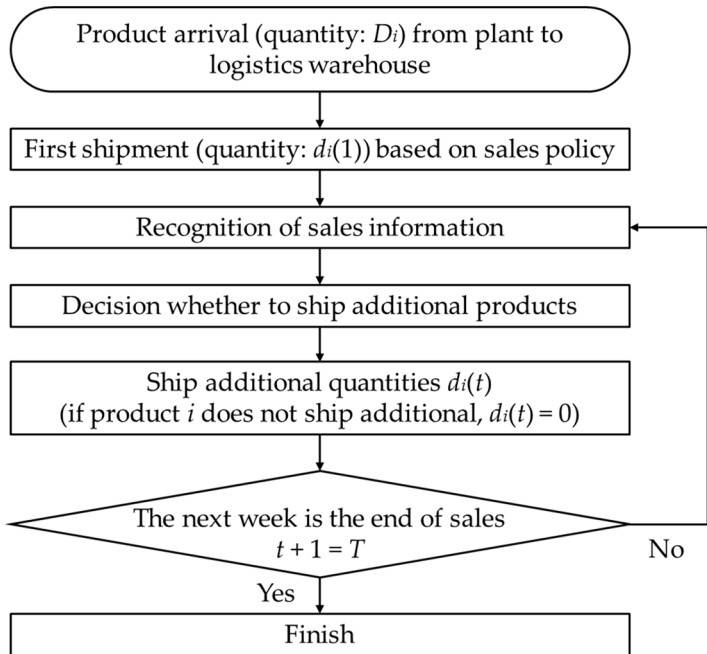

**Figure 4.** Decision-Making Process for Shipping Personnel.

These steps are for product *i*.

Step 1. Product *i* arrives at the logistics warehouse from a plant, and its quantity is $D_i$.

Step 2. Shipping personnel practice first shipment of product *i* based on a sales policy which is given by manufactures and its quantity is $d_i$ (1).

Step 3. Shipping personnel recognize sales information of product *i* for one week.

Step 4. Shipping personnel consider and decide whether to ship additional product during the next week. At this time, they refer not only to sales quantity of product *i* but also to sales information of similar products sold last year and current trends.

Step 5. Based on shipping personnel decision, product *i* is shipped whose quantity is $d_i$ (*t*) at *t*-th week. In the determination of the quantity, the inventory of the store and the scale of the store are taken into consideration. Then, if personnel decide not to make an additional shipment, shipping quantity of the product $d_i$ (*t*) is zero.

Step 6. If the next week, *t* + 1, is end of sales period *T*, go to Step 7. Otherwise, shipping personnel return to Step 3 and repeat from Step 3 to Step 5 until they fulfill Step 6.

Step 7. Shipping personnel finish their decision.

### 4.3. Product Data Extraction

　　Products in the apparel and textile industry come in many categories, colors, and sizes; thus, the amount of available product data enormous. Further, customer demand is uncertain, and sales area affected by fashion and other industry trends that are in a state of constant flux. In light of these circumstances, the data that shipping personnel have access to at any given time is limited in scope. Therefore, when analyzing our own data, we divide the products into several groups, extract data from products likely to require shipping plan revision, and analyze them.

　　An example of the actual change in inventory in a logistics warehouse is shown in Table 1. There is a significant difference in product quantities even in the first week, making it difficult to see inventory transitions of all products on one scale of inventory quantity. Therefore, the shipping ratio is calculated as the incoming inventory in each week $d_i(t)$, normalized by the total arrival quantity $D_i$ of each product $i$:

$$u_i(t) = \frac{d_i(t)}{D_i}. \tag{1}$$

　　After calculating the shipping rate for each product, a $k$-means cluster analysis of $u_i(t)$ is conducted for $t = 1, \ldots, 9$, with a sample of $N = 796$ in order to divide products into groups. Since the number of pieces of data is as large as $N = 796$, we decided to use nonhierarchical instead of hierarchical clustering. To calculate the distance between samples, Euclidean distance is used. One disadvantage of the $k$-means method is the dependence on the initial value. Therefore, some initial values are set and cluster analysis is performed, and initial values are chosen to minimize the distance within the average cluster. The results of the cluster analysis leads to patterns, here called sales policies (i.e., policies involving selling by advertising products that will be recognized by customers vs. policies to sell normally, as standard products).

**Table 1.** Examples of Changes in Inventory.

| | Change in Inventory in Each Week | | | | | | | | | |
|---|---|---|---|---|---|---|---|---|---|---|
| Product | 0 | 1 | 2 | 3 | 4 | 5 | 6 | 7 | 8 | 9 |
| A | 2884 | 291 | 266 | 240 | 198 | 154 | 123 | 80 | 44 | 36 |
| B | 2078 | 1151 | 1038 | 347 | 221 | 171 | 122 | 0 | 0 | 0 |
| C | 293 | 249 | 248 | 220 | 220 | 220 | 191 | 116 | 94 | 32 |

　　The result of the $k$-means method suggested that it was reasonable to divide the analyzed data into two groups, which were then roughly divided into five groups to provide more detail. Figure 5 shows the average value of inventory ratios in each week for each of the five clusters. Depending on the shipment percentages in the first week, five clusters can be broadly categorized.

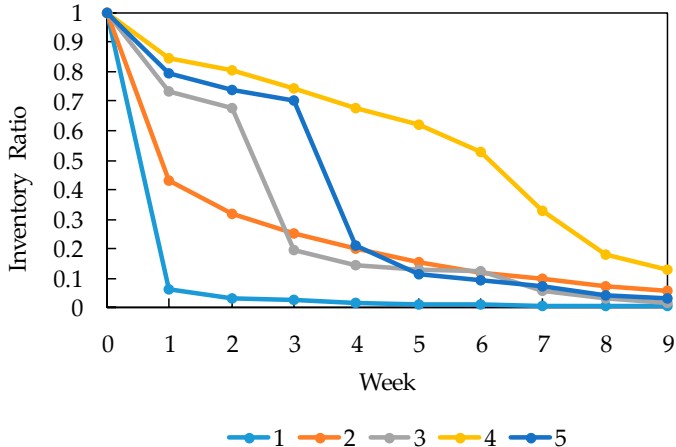

**Figure 5.** Average Inventory Ratios in Each Week for Each Cluster.

For clusters 1 and 2, the shipment ratio in the first week is high. On the other hand, in clusters 3, 4, and 5, the shipment ratio of the first week is low. Investigating the characteristics of each cluster further, products of clusters 1 and 2 represent the policy that a manufacturer wants to sell featured products. Also, the trend of shipment ratios from the second to the ninth week shows that products remaining in the logistics warehouse are gradually shipped until the end of the sale period. Therefore, the shipping personnel will not revise the shipping plan later. On the contrary, products in clusters 3, 4, and 5 demonstrate a policy whereby the shipping personnel sell the product in stages, conditional on the level of sales. Accordingly, products belonging to these classes are highly likely to have been subject to revised shipping plans. In other words, shipping personnel should have recognized products exhibiting this kind of sales information.

### 4.4. Forecasting Inventory Ratios

The cluster analysis identifies products that shipping personnel should recognize as undergoing significant changes in inventory. However, the shipping personnel do not know the optimal time or other factors about the shipment. Therefore, the reference lines obtained by cluster analysis (Figure 5) predict the inventory ratio at certain weeks. Changes in inventory that deviate significantly from the basic line are signals that should be transmitted to shipping personnel to highlight important sales information.

In this paper, to predict how far the inventory transition will be from the basic line, the conditional probability of the bivariate normal distribution is used for each product. In this way, it is possible to predict the subsequent inventory transition when the change in inventory is recognized until a specific week. The conditional expectation and variance in the conditional probability of the bivariate normal distribution are obtained by the following Equations (2) and (3), respectively.

$$\mu' = \mu_Y + \rho \sigma_Y \frac{I_{i,t} - \mu_X}{\sigma_x}, \tag{2}$$

$$\sigma' = \sigma_Y \sqrt{1 - \rho^2} (\leq \sigma_Y). \tag{3}$$

Based on the above conditional expectation and the conditional variance of the conditional probability, a 95% confidence interval is generated, which is used as the range of prediction. However, since the actual inventory ratio may be smaller than the inventory ratio considering 95% confidence intervals for the conditional expectation obtained by the conditional probability of the bivariate normal distribution, this paper proposes minimum values for each week. Since the inventory ratio cannot be smaller than zero, it is considered to be the minimum. Hence, the upper and lower limits of the range from the inventory ratio considering the 95% confidence interval and the actual inventory ratio are as follows:

$$\text{Upper limit}: \ \text{Min}\{\mu' + 1.96\sigma', I_{i,t}\}, \tag{4}$$

$$\text{Lower limit}: \ \text{Max}\{\mu' - 1.96\sigma', 0\}. \tag{5}$$

Here, we predict the inventory ratio of product A using information on all clusters of products to be sold in stages, namely, clusters 3, 4, and 5, and for the cluster to which it belongs, 3. The prediction for product A is shown below. The gray line of each figure is a line connecting the conditional expectation values at each week calculated by the Equation (2). The blue line represents change of the upper limit value in each week obtained from Equation (4). The orange line shows change of the lower limit derived from Equation (5). Finally, in order to compare how the actual inventory ratio changes to changes of gray, blue, orange lines, the change of the actual inventory ratio is indicated by yellow line. Here, the calculation results are shown assuming that the sales period is $T = 9$. Therefore, based on the assumption, the products will not be sold after nine weeks and will be discarded.

The shipment of the first week is determined by the manufacturer's sales policy, and shippers who oversee shipping decisions begin shipping after the second week. Therefore, this paper predicts inventory levels for all weeks, conditional on the realized level of sales up to any given week. Form Figure 6a,b, since the shipping ratio at the third week is large, the range of prediction after the fourth week can be narrowed significantly. Furthermore, with the expected value of the conditional probability, which is the gray line, at the time when the inventory ratio up to the second week is known, the inventory ratio of the ninth week has already come close to the actual inventory ratio. Therefore, in the case of product A, the inventory ratio that will result in disposal could be predicted at the second week. Shipping personnel who know this can then reduce the amount of product disposal by considering various countermeasures and taking action.

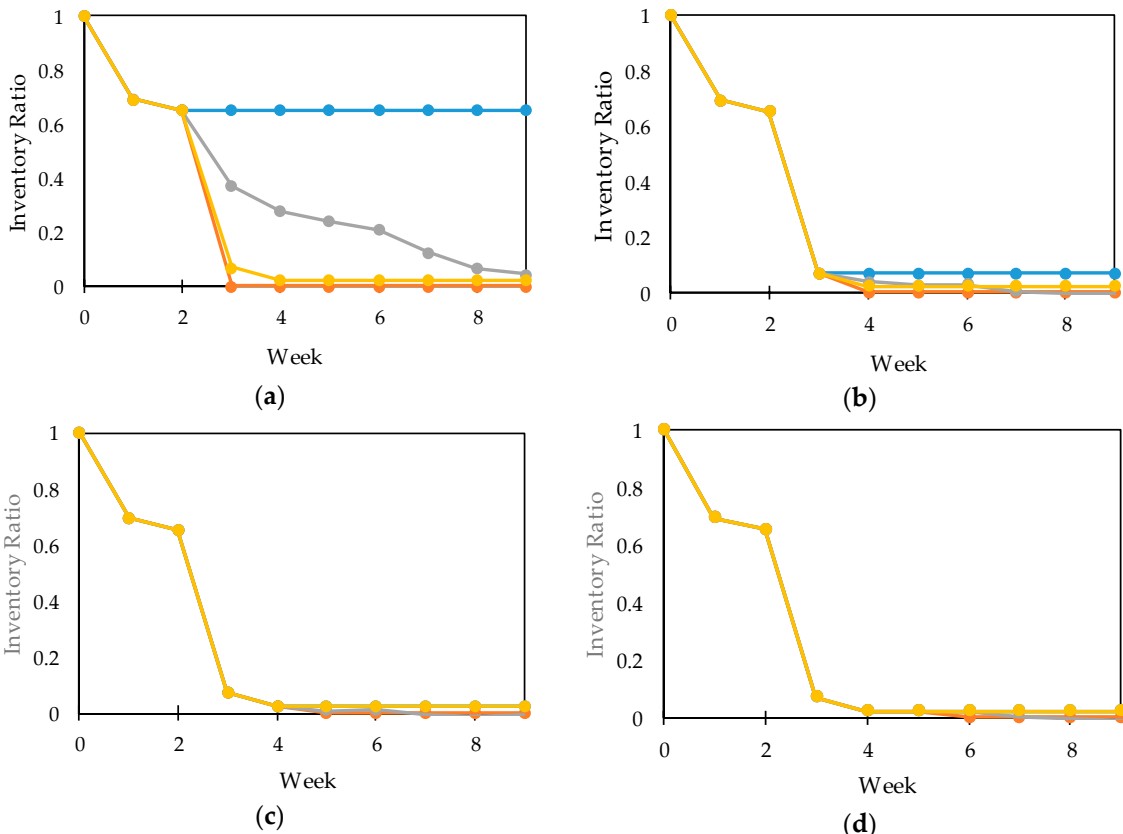

**Figure 6.** The Expectation for Product A (conditional on data of clusters 3, 4, and 5). Note: graph (**a**) represents when the actual inventory ratio is known by the second week, (**b**) is when it is known by the third week, (**c**) is when it is known by the fourth week, and (**d**) is when it is known by the fifth week. The gray line shows the conditional expectation, the blue and orange lines show the upper and lower limits of the range, respectively, and the yellow line shows the change in the actual inventory ratio of product A.

In both Figures 6 and 7, the more weeks in which the inventory ratio is known, the narrower the range of prediction. In the case of product A, the prediction range is narrowed significantly in the third week. This is because the actual inventory ratio of product A is drastically reduced in the third week. Moreover, when predicting with only the cluster to which product A belongs, the prediction range is narrowed as a whole. However, product A is shipped at a high rate in the third week. Then, the inventory ratio at the third week is much smaller than the expected inventory ratio of the third week at the second week. The actual sales are not known at that time, but shipping more than the expected value may result in more waste. Hence, excess shipping can be reduced by shipping the product in the fourth week after receiving more detailed sales information. In other words, since this product is highly likely to result in some disposal, it is possible to reduce extra shipping by lengthening

the period for recognizing sales information. As in the other method, when shipping at the third week, it is better to adjust the shipping amount of product A. For example, product A is shipped at proportions according to the basic line (gray line of Figure 6 or Figure 7).

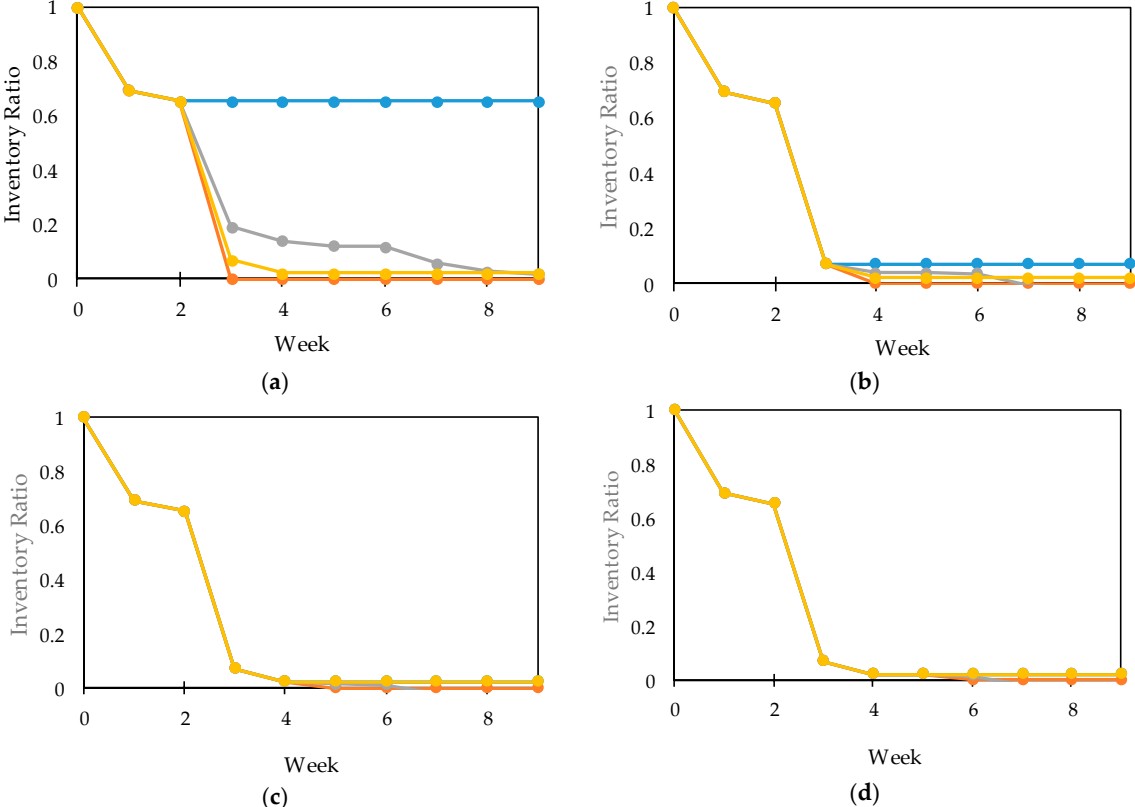

**Figure 7.** The Conditional Probability Expectation of the Bivariate Normal Distribution of Product A for Data of Cluster 3. Note: graph (**a**) represents when the actual inventory ratio is known by the second week, (**b**) is when it is known by the third week, (**c**) is when it is known by the fourth week, and (**d**) is when it is known by the 5th week. The gray line shows the conditional expectation, the blue and orange lines show the upper and lower limits of the range, respectively, and the yellow line shows the change in the actual inventory ratio of product A.

As a second example, the prediction results for product B are shown in Figures 8 and 9. Product B is also sold in stages, and the detailed cluster number is 5. As with product A, we predict cases for which the actual inventory ratio is known in the second week. The figures show that product B has a high inventory ratio even at the ninth week. Also, in the case of product B, the predicted ninth week's inventory ratio at the second week does not differ much from the prediction at the fifth week. Therefore, some degree of prediction is possible in the second week. Furthermore, as can be seen in Figure 9d, the inventory ratio at the sixth week is lower than the expected value, suggesting that shipments were made to avoid being left with a large inventory in the logistics warehouse. However, since the product is unlikely to sell much after being shipped in the sixth week, perhaps it should have been shipped earlier to increase sales. By shipping in this manner, the product would have been placed in the store for a longer period, and opportunities to sell at the fixed price would also have increased. This would have reduced the amount of waste due to unsold product.

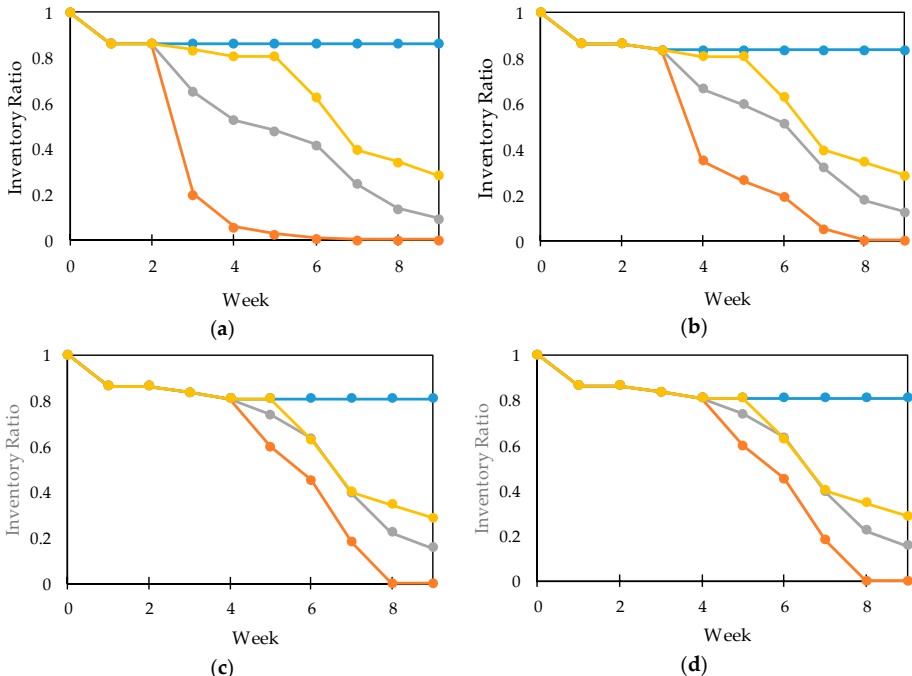

**Figure 8.** The Expectation of Product B (conditional on data for clusters 3, 4, and 5). Note: graph (**a**) represents when the actual inventory ratio is known by the second week, (**b**) is when it is known by the third week, (**c**) is when it is known by the fourth week, and (**d**) is when it is known by the fifth week. The gray line shows the conditional expectation, the blue and orange lines show the upper and lower limits of the range, respectively, and the yellow line shows the change in the actual inventory ratio of product B.

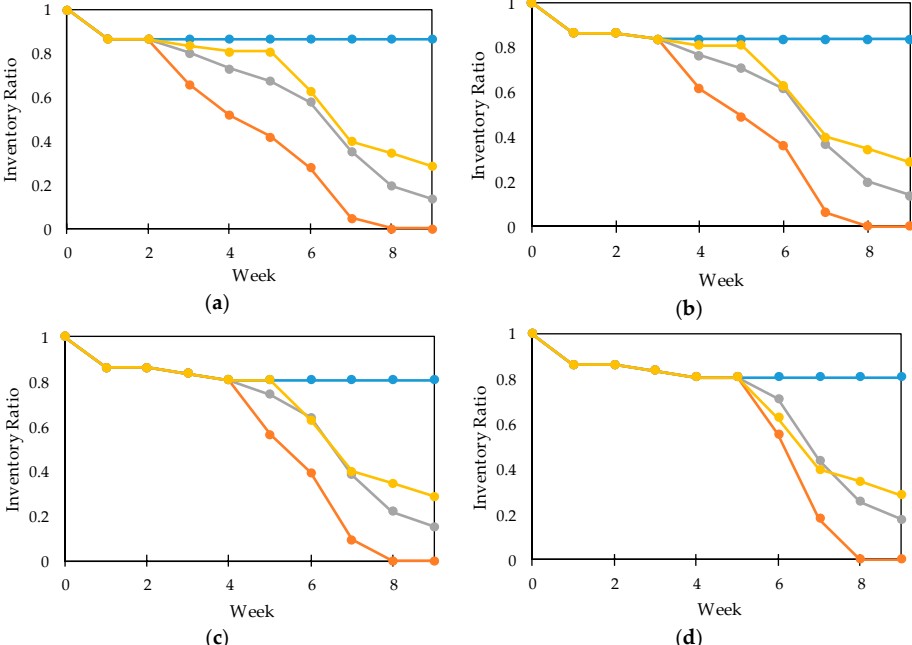

**Figure 9.** The Conditional Probability Expectation of the Bivariate Normal Distribution of Product A for Data of Cluster 5. Note: graph (**a**) represents when the actual inventory ratio is known by the second week, (**b**) is when it is known by the third week, (**c**) is when it is known by the fourth week, and (**d**) is when it is known by the fifth week. The gray line shows the conditional expectation, the blue and orange lines show the upper and lower limits of the range, respectively, and the yellow line shows the change in the actual inventory ratio of product A.

When using the results of prediction, first, the shipping personnel would compare the conditional expectation with the inventory ratio of the product at the present moment. This shows how the inventory ratio of the product changes with respect to the average. Then, the shipping personnel would plan the shipping for the next week. Finally, they would determine the amounts of the shipment while also taking into account the sales policy.

Also, the results of this prediction lead to different recommendations depending on the type of product. For example, in the case of a standard product, manufacturers sell for a longer term, so if the inventory runs out earlier than forecasted, it is possible to produce additional products to meet customer demand. On the other hand, in the case of a product sold for only one season, when the expected value of the stock ratio at the ninth week is large, this is a signal that products are not selling well that should be heeded by shipping personnel.

In QR strategy in the logistics warehouse, product shipping quantities are decided at the time when shipping personnel receive sales information on each product during the monitoring period. However, in this paper, we give a signal to shipping personnel that is based on the comparison of previous inventory transition for a similar product to the current inventory transition for the current product. Therefore, even if there is no arrival of sales information, shipping personnel can receive signals about shipping. In addition, this method is effective for all sales periods.

## 5. Conclusions

In this study, based on the premise of selling at fixed price, we tested an approach for reducing waste from unsold products in the textile and apparel industry and, thereby, mitigating adverse effects on the environment. The purpose of shipping personnel in the study is to recognize relevant data about sales situations and make appropriate judgments (i.e., predictions) about whether to issue alerts. The alerts in this paper are made from the result of data analysis to the product that shipping personnel should watch sales situation carefully. Although there are many kinds of data that can be analyzed in this particular industry, we chose to focus on logistics warehouse data, examining stock points to assess changes in the warehouse inventory ratios. Since the total quantities of apparel products varied considerably, the data was normalized in our study. Furthermore, because it was difficult to monitor the status of each individual product, we categorized products into several groups using cluster analysis. We found that the products whose sales information required the most careful monitoring were products that were sold in stages. For these products, we also forecasted inventory ratios using cluster results, and predictions showed that some products were more likely to remain unsold. Given this, we considered the products whose shipment patterns (i.e., timing and quantities) should be modified.

This study has practical value for shipping personnel. By using conditional probability in the manner we have proposed, shipping personnel could accurately predict inventory ratios at the ninth week from changes in the inventory ratios up to the third week. Therefore, at the third week, shipping personnel could take remedial measures for products predicted to have excess inventory in the logistics warehouse.

In light of the trade-off between the risk of disposal from unsold product and the risk of opportunity loss from product shortage, future research should investigate optimal shipment timing when disposal and shortage are simultaneously taken into consideration. Also, the optimal shipping quantities should be obtained at the optimum shipment timing. In practice, however, the timing or quantities of products shortage cannot be obtained as data. Therefore, under the assumption of a product shortage situation, the risk of opportunity loss will be evaluated. In addition, future research should verify the effectiveness of the proposed method by performing analyses in several different scenarios. In sum, using past data as well as current sales data can allow for better detection of market trends. Although the larger the amount of data, the easier it is to detect market trends in past and present, to do so, many textile and apparel companies are needed to share and analyze data. Apparel and textile companies can use our findings to improve the timing and quantity of shipments,

more effectively manage supply chains, and protect profits. In so doing, they can also reduce the burden of disposal, minimize adverse effects on the environment, and contribute to the development of more sustainable societies.

**Author Contributions:** R.T. performed the research and wrote the manuscript. A.I. coordinated and supervised the overall research. T.S. assisted in developing mathematical methods. M.H. and W.K. acquired funds and collected data. All authors read and approved the final manuscript.

**Conflicts of Interest:** The authors declare no conflict of interest.

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
