# Peer review of "Data Analysis of Shipment for Textiles and Apparel from Logistics Warehouse to Store Considering Disposal Risk"

_sustainability, doi:10.3390/su11010259_

Round 1

Reviewer 1 Report

1. in this study, on the premise of selling at a fixed price, we consider the timing of shipping as a method to reduce waste risk due to unsold inventory. However, in practice, the discount should be considered. 2. The literature review should be improved sinced it only cover the quick response. 3. In this research, k-means cluster analysis is used. Why K-mean is used? The author should clairfy and explain it. 4. The model is too simple. Therefore, the contribution is not significant.

Author Response

Response to Reviewer 1 Comments

We thank reviewer for careful reading our paper and for giving constructive comments. Based on your comments, modifications have been made in a revised paper. A summary of the modifications is as follows:

Point 1: In this study, on the premise of selling at a fixed price, we consider the timing of shipping as a method to reduce waste risk due to unsold inventory. However, in practice, the discount should be considered.

Response 1: In actual sales, we need to consider discounts. However, in order to think on the premise of fixed price, we define the period that can be sold at regular price as the sales period. This period is nine weeks. Therefore, we added prerequisites.

Point 2: The literature review should be improved since it only cover the quick response.

Response 2: Because there was not enough literature review, we conducted a literature review on sustainable, textile and apparel supply chain and forecast in textile and apparel industry and we added the results.

Point 3: In this research, k-means cluster analysis is used. Why K-mean is used? The author should clairfy and explain it.

Response 3: Since the reason for using the k-means method was not clear, we added the reason for using that method. It is written in “4.3. Product Data Extraction”.

Point 4: The model is too simple. Therefore, the contribution is not significant.

Response 4: The contribution of our research was not clear. In this research, we restrict the data to be seen in the textile and apparel industry which has many data. This makes it possible to assist shipper’s personnel in making decisions. Also, as the number of products sold at regular price increases, there is aim to improve product circulation. Therefore, we added our consideration of the results and modified the expression in the conclusion.

Reviewer 2 Report

Thank you for the possibility of the paper review.

The paper seems to be interesting with well written introduction and paper aim presentation based on the literature gap discussion. The methods are properly presented. The study is widely described.

The paper has numerous imperfections those should be improved:

There is lack of methodology as well as main results description in the Abstract.

Literature review is disappointing, very narrow and not presenting suitable background for further study.

There is no results discussion also no comparision to previous study.

Conclusion is narrow, very limited recommendations as well as no study limitations presented.

Author Response

We thank reviewer for careful reading our paper and for giving constructive comments. Based on your comments, modifications have been made in a revised paper. A summary of the modifications is as follows:

Point 1: There is lack of methodology as well as main results description in the Abstract.

Response 1: We corrected the abstract. We wrote in detail about the results and methodology.

Point 2: Literature review is disappointing, very narrow and not presenting suitable background for further study.

Response 2: Because there was not enough literature review, we conducted a literature review on sustainable, textile and apparel supply chain and forecast in textile and apparel industry and we added the results.

Point 3: There is no results discussion also no comparision to previous study.

Response 3: Due to insufficient discussion of our results, we added discussion to our results mainly compared with the Quick Response (QR) strategy.

Point 4: Conclusion is narrow, very limited recommendations as well as no study limitations presented.

Response 4: In the conclusion, we changed to sentences to summarize our study. We added detailed sentences on research motivation, method, result, discussion.

Reviewer 3 Report

I am not an expert with quantitative research methods, thus the comments are towards other parts and I leave other reviewers to provide comments on the mathematics.

Overall, the paper needs proof reading for English. The literature review section needs to be strengthened with a review on existing studies, some of the descriptions in the introduction can be moved to this part. For me the research framework is too ideal and simple with only one warehouse and one store. Normally in practice there are multiple stores and the stocks could be relocate to other stores. Please provide the reasons to the model and assumptions. Finally, the practical implications should be provided to highlight its practical contributions.

Author Response

We thank reviewer for careful reading our paper and for giving constructive comments. Based on your comments, modifications have been made in a revised paper. A summary of the modifications is as follows:

Point 1: The literature review section needs to be strengthened with a review on existing studies, some of the descriptions in the introduction can be moved to this part.

Response 1: We moved some sentences in introduction to the chapter on literature review. Because there was not enough literature review, we conducted a literature review on sustainable, textile and apparel supply chain and forecast in textile and apparel industry and we added the results.

Point 2: For me the research framework is too ideal and simple with only one warehouse and one store. Normally in practice there are multiple stores and the stocks could be relocate to other stores. Please provide the reasons to the model and assumptions.

Response 2: We added some explanations to “Survey on State” on the process that we used such a model. Although it is an apparel supply chain which originally has a complicated structure, we are planning to grasp the situation simply by looking at the basic data.

Point 3: Finally, the practical implications should be provided to highlight its practical contributions.

Response 3: It was unclear that our results influence shipper’s personnel decision making. For that reason, we added sentences of discussion for the results. This will provide our contribution.

Reviewer 4 Report

I was pleased to read your study and I believe it raises interesting insights. The literature section brings robust arguments based on references that support the hypotheses development. Despite the paper is well presented in its present form, a few improvements are suggested as follow:

Q1. More details on the methodology could be provided. A more comprehensive description summarizing the phases of analysis or a more specific description of methodology will help the readers to understand the different methodological steps.

Q2. The literature review section could be improved including recent contributions in the body of literature. When deepening you theoretical argumentation you may, in every section, start from a general standpoint before focusing on the specific context of investigation.

Q3. With regard to the main results, could the context of investigation affect the results? Why? Please stress managerial implications of the study. Future research opportunities should also be better emphasized.

Q4. The quality of figure 1 should be improved.

Author Response

We thank reviewer for careful reading our paper and for giving constructive comments. Based on your comments, modifications have been made in a revised paper. A summary of the modifications is as follows:

Point 1: More details on the methodology could be provided. A more comprehensive description summarizing the phases of analysis or a more specific description of methodology will help the readers to understand the different methodological steps.

Response 1: We added sentences because there was a lack of description about the methodology of analysis we conducted. Specifically, we increased the explanation for the “Chapter 4. Methods”.

Point 2: The literature review section could be improved including recent contributions in the body of literature. When deepening you theoretical argumentation you may, in every section, start from a general standpoint before focusing on the specific context of investigation.

Response 2: Because there was not enough literature review, we conducted a literature review on sustainable, textile and apparel supply chain and forecast in textile and apparel industry and we added the results of literature review.

Point 3: With regard to the main results, could the context of investigation affect the results? Why? Please stress managerial implications of the study. Future research opportunities should also be better emphasized.

Response 3: There was insufficient discussion on our results, and we did not mention the impact on shipper's personnel decision making. For that reason, we added some discussion for the results. We also added explanations on how it will affect the management of the company.

Point 4: The quality of figure 1 should be improved.

Response 4: Figure 1 is an image of the apparel supply chain we deal with. However, because the flow of the product was not clearly expressed, we changed the Figure 1.

Round 2

Reviewer 1 Report

All comments are addressed.  Here are few suggestions.

For Figure 6-9, I will suggest to describe the meaning of different line color.  

For Figure 6-9, the inventory ratio is almost 0.  Is it reasonable?

Author Response

We thank reviewer for careful reading our paper and for giving constructive comments. Please check the pdf file for details.

Reviewer 2 Report

Thank you for improvements. Some check is still necesarry eg:

authors used name in reference in the text "Sébastien (2010)" while they should use surname of the author (see the reference list) ! Please check 

Author Response

(The authors gave the same response as above.)

Reviewer 3 Report

I am happy with the changes.

Author Response

We thank reviewer for careful reading our paper. We took a native speaker check of English to proofread our English writing.